# FRP Confinement of Stone Samples after Real Fire Exposure

**DOI:** 10.3390/polym12102367

**Published:** 2020-10-15

**Authors:** Luis Estevan, F. Javier Baeza, Francisco B. Varona, Salvador Ivorra

**Affiliations:** Department of Civil Engineering, University of Alicante, P.O. Box 99, 03080 Alicante, Spain; luis.estevan@ua.es (L.E.); fj.baeza@ua.es (F.J.B.); borja.varona@ua.es (F.B.V.)

**Keywords:** stone, masonry, confinement, FRP, fire, high temperature

## Abstract

The mechanical properties of stone materials can be severely affected by exposure to high temperatures. The effect of fire on stone buildings could cause irreversible damage and make it necessary to retrofit the affected elements. Particularly, the strengthening of columns by confinement with composites has been widely improved during the last decades. Today, fiber reinforced polymer (FRP) confinement represents a very interesting alternative to traditional steel solutions. This work studied the behavior of cylindrical stone specimens subjected to real fire action and confined by means of CFRP or GFRP jackets, with the aim of assessing the effectiveness of these reinforcement systems applied to a material that has previously been seriously damaged by high temperature exposure. In general, the strengthened samples showed notable increases in strength and ductility. The response seemed to depend basically on the FRP properties and not on the degree of damage that the stone core may have suffered. Finally, the results obtained experimentally were compared with the confinement models proposed by the available design guides, in order to evaluate the accuracy that these models can offer under the different situations addressed in this research.

## 1. Introduction

Stone is generally considered as one of the most robust building materials in terms of fire resistance. However, recent episodes, such as the fires in Notre-Dame Cathedral in Paris or the National Museum of Brazil in Rio de Janeiro, showed that the action of fire in historic buildings can have catastrophic consequences and jeopardize the safety of people and the integrity of invaluable heritage. The alteration of the physical, chemical, and mechanical properties of stone materials subjected to high temperatures has been the subject of numerous studies in recent decades. Brotons et al. [1] worked with samples of calcarenite that had been subjected to temperature levels between 200 and 600 °C in an electric furnace, letting them cool slowly to room temperature in the laboratory or cooling them abruptly by immersion in cold water. The results showed significant drops in the compressive strength of the rock (up to 34% in the slowly cooled samples and 53% in the water cooled ones, for 600 °C exposure) and, especially, in the modulus of elasticity, with reductions up to 85% (with both cooling systems). Kumari et al. [2] proposed a similar experimental study with granite samples exposed to temperatures between 100 and 800 °C, with slow and sudden cooling, obtaining strength decreases of up to 80% and drops in the modulus of elasticity of around 90%, somewhat greater in the series subjected to quick cooling. In other investigations published with different stone materials [3,4,5], similar results can be observed, and, in general terms, the mechanical properties of rocks were seriously affected by exposure to high temperatures.

Although the alteration of the properties of rocks exposed to high temperatures has been widely studied during the last years, in most cases, the work is done under ideal laboratory conditions using electric furnaces, while investigations with exposure to real fire are scarce and usually limited to the analysis of historical buildings affected by fires [6]. It is clear that the use of electric furnaces in the laboratory allows the simplification and standardization of tests in order to facilitate the comparison of results with those of other investigations [7]. However, the atmosphere created inside the furnace does not reflect the real conditions of a building during the fire event, largely because the main mechanism of heat transmission is convection in the former case and radiation in the latter [8]. On the other hand, the controlled heat gradients generated in the furnace have little to do with the effect of a real fire, where the temperature distributions are variable and fluctuate rapidly; besides, smoke and ash produced during combustion can also have an impact on the behavior of the material [9].

Chakrabarti et al. [10] studied the effects of fire on stone materials in some historic buildings in the UK affected by fire. One example is St. Michael’s Church (Newquay), where the fire lasted for three hours and it was estimated that the stone may have reached temperatures in the range of 300 °C, causing crumbling of up to 3 cm deep in some columns, which later forced the complete replacement of these elements with new material. Hajpál [11] analyzed some historical buildings in Hungary and Germany affected by fires, and found that the stone elements suffered variable degrees of damage due to the temperature gradients that occur between exposed and protected areas or between the slenderest or most massive pieces, situations that are difficult to simulate in laboratory tests with rock samples and electric furnaces. Koca et al. [12] studied two marble columns after the fire at Mithatpasa Technical Junior High School in the city of Izmir, Turkey, which lasted two hours and produced temperatures in the order of 700 °C inside the building. The mechanical behavior of the affected stone was analyzed by extracting samples that were tested in the laboratory, obtaining strength losses of 25%. Some samples were re-exposed to different temperature levels, and it was found that the material would disintegrate at temperatures above 300 °C. This would show that the stone, seriously damaged after a first fire, could hardly withstand the action of a fire again.

The main conclusion that can be drawn from all these investigations is that the effect of fire on stone buildings can produce severe and irreversible damage and make it necessary to retrofit the affected elements. In the particular case of the columns, the reinforcements have traditionally been made by means of steel jackets, although along the last decades the technique of confinement with composite materials has been notably developed, usually by wrapping with fiber reinforced polymer (FRP) sheets. The behavior of concrete confined with composite materials has been widely studied since the 1990s [13,14,15], and theoretical models allow estimating the mechanical performance of strengthened elements with high accuracy [16,17]. Nevertheless, research with stone or masonry elements is much scarcer for the moment.

Aiello et al. [18,19] tested small columns made of limestone blocks and CFRP confinement either continuously or by means of horizontal stripes with different width and separation, obtaining strength increases of up to 93% and ultimate deformations by almost 200% higher. It is interesting to see how discrete jacketing was almost as effective as continuous wrapping, a matter of great importance in case of columns exposed to moisture, where this solution would allow the stone core transpiration. Regarding the confinement of square cross-section pieces, it was found that the efficiency of the jacket was lower due to stress concentration at the corners, which could be improved by increasing the radius of curvature of the samples. Faella et al. [20] proposed a study with square cross-section specimens made of limestone or volcanic tuff blocks and confined with CFRP or GFRP continuous jackets. In the case of specimens with a greater radius of curvature at the edges and with two layers of reinforcement, the ultimate strength was doubled, with large increases in the ultimate strain. Micelli et al. [21] presented a similar investigation to [18], in which they also incorporated hybrid solutions made of glass fiber fabrics combined with shape memory alloys (SMA), in order to evaluate the capacity of these materials to provide active confinements. Witzany and Zigler [22] worked with larger square cross-section columns made of perfectly cut sandstone pieces or in the form of irregular blocks and continuous or discontinuous CFRP reinforcements. Estevan et al. [23] proposed an experimental investigation with cylindrical specimens formed by calcarenite samples similar to those presented in [1] and different strengthening systems: FRP with unidirectional or multidirectional carbon or glass fabrics and FRCM (fabric reinforced cementitious matrix) with glass or basalt meshes. The results clearly showed a higher efficiency of the FRP confinement, doubling the ultimate strength with respect to the unreinforced samples and increasing the ultimate strain by more than 10 times in some series. With respect to the FRCM strengthening, a smaller confinement capacity was observed, obtaining strength increases of just 26% in the best of cases. In a subsequent study with the same type of rock, Estevan et al. [24] studied the strengthening of samples subjected to moisture and preload with CFRP and GFRP jackets. In general, the response of a retrofitted element seemed to depend fundamentally on the properties of the FRP sheet and not on the degree of damage that the stone core might have previously suffered.

Despite all the research presented above, the studies on structural elements retrofitted with FRP confinement under compression after high temperature exposure is quite limited at the moment. Some works have dealt with concrete specimens subjected to different temperature ranges under laboratory conditions, i.e., using electric furnaces, and wrapping with CFRP [25,26]. Nonetheless, publications about stone or masonry elements are practically non-existent. Estevan et al. [27,28] studied the behavior of calcarenite samples exposed to temperatures of up to 600 °C in an electric furnace with slow and quick cooling. The stone suffered notable losses of strength (around 40–50%) and a severe reduction of its modulus of elasticity (over 80%). Subsequent confinement was performed wrapping with CFRP and GFRP (one layer), and the results in the damaged specimens showed a behavior very similar to undamaged rock samples with equivalent reinforcements. In other words, and in line with the results obtained in [24], it can be concluded that the mechanical performance of the structural elements retrofitted by FRP confinement seems to depend basically on the properties of the composite sheet and not on any previous alteration suffered by the stone material.

As a summary of the research presented throughout this introduction, two main conclusions may be highlighted. First, the mechanical properties of stone materials subjected to elevated temperatures can be severely affected, although studies with exposure to real fire are very scarce at present. Second, FRP confinement of structural members subjected to compression can be a very effective solution in many cases, although, in the particular case of the strengthening of stone members damaged by fire, hardly any previous studies are available in scientific literature. Therefore, the main objective of this study aimed to expand with experimental results the state of knowledge in this field. In addition, the current structural codes for masonry reinforcement do not include any comments regarding the confinement of damaged materials. In fact, the efficiency of the FRP itself is a key parameter to assess the strength gain of FRP confined columns, and it has been determined based on experimental data. There is no consensus about its value for different fibers; hence, there is a great uncertainty between codes. Therefore, a secondary objective was aimed at the evaluation of the current structural codes for the reinforcement of masonry structures, in which mechanical properties had been damaged by fire exposure. The experimental data will be compared with the values obtained with the formulation given in the standards in order to check the accuracy of the prediction of the strength gain in the case of FRP jackets for columns confinement.

## 2. Materials and Methods

### 2.1. Materials, Experimental Program and Test Setup

The stone material used in this research was a calcarenite known locally as Piedra de San Julián, from a rock mass located by the sea northeast of the city of Alicante (Spain). This rock served as base material for some of the previous studies presented in the state-of-the-art review [1,23,24,27,28]. The samples were collected from several geotechnical surveys carried out during the construction of a railway tunnel through the mountain. A total of 50 cylindrical samples were prepared, 72 mm diameter and 180 mm height, in order to keep the ratio 1:2.5 recommended by ASTM D7012-14e1 standard [29]. The main properties of the rock were (average ± standard deviation): bulk density 1982 ± 28 kg/m^3^; compressive strength 19.99 ± 1.61 N/mm^2^; modulus of elasticity 11859 ± 475 N/mm^2^; Poisson’s ratio 0.22 ± 0.03.

The retrofitting of the samples was made with CFRP and GFRP jacketing (one layer). For determining the mechanical properties of the composites, 5 samples were prepared with each type of fabric, with dimensions 25 × 250 mm, which were tested in tension in accordance with the procedure defined in ASTM D7565/D7565M-10(2017) standard [30]. The tests were carried out in an electrical press equipped with a load cell of 20 kN capacity; the deformation was registered with a LVDT device located at the midpoint of the specimen. Table 1 summarizes the main properties of the materials: in case of fabrics and epoxy resin, the data provided by the manufacturer are indicated; for composite materials, the results obtained experimentally are reported (average of five specimens tested in each series and coefficient of variation, in brackets).

The procedure for applying the FRP confinement to the stone specimens can be summarized in the following phases. First, cleaning with steel brush and compressed air. Afterwards, a layer of epoxy resin was applied using a brush, evenly distributed on the surface of the specimen. Then, the fabric was placed with fibers aligned with the transverse direction of the sample and an overlap length of 25% (60 mm, determined according to different tests carried out previously). Finally, an aluminum grooved roller was used to apply lateral pressure, while a second layer of resin was added, checking the removal of any crease or occluded air bubbles. All samples were cured at room temperature for at least 7 days, in order to comply with the manufacturer’s specifications.

The tests were scheduled as shown in Table 2, where the specimens have been identified with a five-digit code, A.BB.XX, in which:‘A’ encodes the confinement solution: N (non-confined); C (CFRP); G (GFRP). All FRP jackets comprised only one layer of fabric‘BB’ encodes the treatment that the stone specimen had previously undergone: N (no treatment); FD (real fire exposure with slow dry cooling); FW (real fire exposure and quick cooling by means of water spraying).‘XX’ encodes the specimen’s number within each series (01, 02, 03, or 04).

All samples were tested in uniaxial compression in an electric press equipped with a 300 kN load cell. The lower plate of the press was fixed, while a spherical seat was incorporated in the upper one. The tests were programmed with a constant stress rate of 0.5 MPa/s, until the specimen failed, according to ASTM D7012-14e1 standard [29]. To register the axial strains, a rigid frame was screwed to the lower plate of the press, on which two LVDTs were installed to measure the descent of the upper plate. Transverse strains were measured by means of two strain gauges per specimen, located at mid-height and diametrically opposite points. LVDTs are Novotechnik TEX Series position transducers, with a measurement range of 50 mm; the strain gauges are linear HBM LY Series, with a measuring length of 20 mm and a nominal resistance of 120 Ω. All tests were recorded with a HBM Spider data acquisition equipment, programmed at 2 Hz frequency. Figure 1 shows a detail of the press used in the tests and the deformation measurement devices.

### 2.2. Fire Exposure

The thermal treatment of the stone samples was carried out in the facilities of the Provincial Fire Brigade Consortium of Alicante (Spain). In a container used for testing and training of the firemen, a real fire was generated by the ignition of combustible material, composed basically of pieces of wood and fiber or particle boards. The amount of material and its placement was determined according to several tests carried out previously. The cabin had inner dimensions of 12 m (length), 2.4 m (width), and 2.4 m (height). The specimens were placed at the top, about 40 cm from the ceiling, suspended by metal chains. For temperature control inside the container, six thermocouples were installed, evenly distributed among the stone samples, and connected to a data acquisition system controlled by a computer located outside. Figure 2 shows the floor plan and section of the container, with the approximate location of the samples and temperature sensors for each fire exposure. 

Two different tests were carried out, in order to evaluate the influence of the extinguishing method in case of a fire in a real building. In the first one (FD test, Table 2, series 04 to 06), the fire was naturally put out, keeping the specimens inside the container for 24 h until an ambient temperature of around 20 °C was reached. In the second test (FW test, Table 2, series 07 to 09), the fire was extinguished when the flames were fully developed (approximately 22 min from ignition) with the assistance of a firemen team, who sprayed the specimens with water until their complete cooling.

Figure 3 shows some relevant pictures of the fire exposure. A general view of the container with the stone samples and the combustible material prepared for ignition is included in Figure 3a. A detail of thermocouples relative location can be seen in Figure 3b. All of the process was made with the assistance of firefighters, including the ignition of combustible material (Figure 3c), the development of the fire inside the cabin (Figure 3d), and the fire extinction (Figure 3e). In addition, a view of the specimens at the instant immediately after being cooled is included in Figure 3f.

Temperature recording curves inside the cabin are shown in Figure 4, for the two tests performed: FD and FW. It should be noted that some temperature sensors did not provide reliable data during the course of the tests, so only those considered valid have been represented (sensors S1, S2, S5, S6 in FD test and sensors S2, S3, S4, S6 in FW test). Both graphs have been drawn with different time scales on the abscissa axis, in order to facilitate the display of curves. As can be seen, the maximum temperature inside the container was reached quickly, approximately 5 min after the ignition of the combustible material. The average temperature with a fully developed fire was around 600 °C, although peaks above 800 °C were registered in some sensors. In the FD test (Figure 4a), a generalized drop in temperature can be appreciated from minutes 20 to 25, which corresponds to the gradual extinction of the flames once most of the combustible material had been consumed. For this reason, it was decided to intervene at the 22nd minute for the rapid cooling of the samples in the FW test, Figure 4b, where EX represents the instant when the spraying with cold water was started with the assistance of the firefighters team.

After the fire exposure tests, some rock samples were cracked or even completely fractured, especially in case of the quickly cooled ones. Once the specimens had been taken back to laboratory, the least damaged were selected and FRP wrapping was done (in 05, 06, 08, and 09 sets), as detailed above. The samples sprayed with water had been previously dried in an electric oven at a temperature of 105 °C for a period of 24 h.

## 3. Results and Discussion

In this section, the results of the uniaxial compression tests corresponding to the series of unconfined samples (series 01, 04, and 07) and those obtained in the confined ones (series 02, 03, 05, 06, 08, and 09) are presented separately. Finally, the results obtained experimentally will be compared with the predictions of the confinement models defined in the available design guidelines.

As explained above, two extensometers measured axial strains, while transverse strains were obtained by means of two strain gauges per specimen. However, the stress-strain curves presented in the following sections, include only axial strains, in order to facilitate their view and interpretation. Nevertheless, the ultimate transverse deformation is required to calculate the FRP strain efficiency factor ($k_{\varepsilon }$), and the values are given in the corresponding tables, as will be explained later. All graphs include the average of four samples in each series as a dark line. In addition, the corresponding bilateral confidence interval, which is shaded in a grey background, was added to evaluate the experimental dispersions. Thus, 95% bidirectional confidence intervals were determined, considering mean values and standard deviations of the stress, for each strain, according to UNE 66040:2003 standard [31].

The following notation has been used:
$f_{\mathrm{mo}}$ ultimate compressive strength of unconfined stone (MPa)$f_{\mathrm{mc}}$ ultimate compressive strength of confined stone (MPa)$\varepsilon_{\mathrm{mo}}$ ultimate axial strain of unconfined stone$\varepsilon_{\mathrm{mc}}$ ultimate axial strain of confined stone$\varepsilon_{\mathrm{mc},t}$ ultimate transverse strain of confined stone$\varepsilon_{\mathrm{fu}}$ ultimate strain of FRP, obtained in direct tensile tests (see Table 1)$k_{\varepsilon }$ FRP strain efficiency factor ($k_{\varepsilon }{=\varepsilon }_{\mathrm{mc},t}/\varepsilon_{\mathrm{fu}}$)$E_{\mathrm{mo}}$ modulus of elasticity of unconfined stone (MPa), determined as the average slope of the straight-line portion of the stress-strain curve, according to ASTM D7012-14e1 [29]$E_{\mathrm{mc}}$ modulus of elasticity of confined stone (MPa), determined as $E_{\mathrm{mo}}$ on the initial branch of the bilinear stress-strain curve$g_{m}$ stone mass-density (kg/m^3^)

### 3.1. Unconfined Samples

The results corresponding to the unconfined samples are summarized in Table 3, and the axial stress-strain curves of the three series tested are shown in Figure 5. As can be seen, the undamaged specimens showed an approximately linear response with brittle failure, a common behavior in this type of stone material. Exposure to real fire produced a notable decrease in the ultimate compressive strength (65% and 70% in the samples with slow and quick cooling, respectively). The ultimate strain increased in both series by more than 4 times with respect to the undamaged stone specimens, which resulted in large reductions of the modulus of elasticity (about 90%), regardless the cooling system used. It is worth noting the great homogeneity of the results obtained, in view of the confidence intervals amplitude, considering that the tests deal with a natural material, such as stone is. As can be seen, the greatest dispersions were obtained in the samples with quick water cooling.

Regarding the type of failure, some examples corresponding to the three series tested are shown in Figure 6. The rock exhibited a brittle fracture, with the specimen cracking in the direction of the load application, regardless of the heat treatment and cooling system used. Exposure to real fire produced a change in the color of the stone material, which darkened and acquired brown and reddish tones.

### 3.2. FRP Confined Samples

The results obtained in the six series of CFRP and GFRP confined stone samples are reported in Table 4, which includes the ultimate compressive strength, ultimate axial strain and the ratios $f_{\mathrm{mc}}/f_{\mathrm{mo}}$ and $\varepsilon_{\mathrm{mc}}/\varepsilon_{\mathrm{mo}}$. These ratios are given in order to evaluate the efficiency of the retrofitting system used. Table 4 also includes the modulus of elasticity of the FRP confined specimens, calculated as the average slope of the straight-line portion on the initial branch of the bilinear stress-strain curve. For the comparison of this modulus with that of unconfined samples, the $E_{\mathrm{mc}}/E_{\mathrm{mo}}$ ratio is also provided. The ultimate transverse strain $\varepsilon_{\mathrm{mc},t}$ was obtained from the average data reported by the pair of strain gauges attached to each specimen. This value allows the calculation of the FRP strain efficiency factor, $k_{\varepsilon }{=\varepsilon }_{\mathrm{mc},t}/\varepsilon_{\mathrm{fu}}$, where $\varepsilon_{\mathrm{fu}}$ represents the ultimate strain of the FRP sheet, obtained experimentally in direct tensile tests with plane laminates samples and included in Table 1. This coefficient takes into account the fact that the $\varepsilon_{\mathrm{fu}}$ strain is never reached in the FRP wrappings, as is reported in some detail in most of the published works and available confinement models. This premature failure of the FRP jacketing is mainly attributed to two reasons: (1) the wrapping is subjected to a triaxial stress state that is not present in plane FRP laminates, tested in uniaxial tensile loading; (2) the microcracking of the substrate generates small local discontinuities that may cause minor stress concentrations that may lead to premature failure of the FRP jacket.

Figure 7 shows the stress-strain curves of the six series of stone samples confined with FRP. The graphs also include in a red dashed line the curves for the equivalent series with unconfined specimens, in order to facilitate the interpretation of the results. In case of undamaged stone specimens, shown in Figure 7a,d, the graphs show the characteristic bilinear response of the FRP confined stone (or concrete) elements that has been reported in most of the published research. In the first phase of the test, the behavior of the reinforced specimen is similar to that of the unconfined stone, since at reduced stress levels, the lateral expansion of the core is negligible, and the effect of the confinement remains practically inactivated. Actually, the slope of the initial branch approximately coincides with the modulus of elasticity of the undamaged stone. This initial branch extends now beyond the ultimate compressive strength of the unconfined stone (about 30 to 35 MPa), as has been observed in other published research with stone materials [18,21,23]. From this point, core cracking occurs, and transverse expansion of the rock becomes much more pronounced; this deformation is limited by the effect of the jacketing, which is rapidly activated. Finally, the curve is stabilized in a second branch, approximately straight and flatter, until the specimen fails, with notable increases in strength and ultimate strain with respect to the unconfined stone material. As can be seen, a slightly higher ultimate strength is obtained in case of CFRP confinement (54.70 MPa) than in GFRP (49.47 MPa), while the ultimate strain is somewhat greater in the latter case. These very small differences are due to the weight of the fabrics used (300 g/m^2^ in case of carbon fiber fabrics and 900 g/m^2^ in case of glass fabrics, as specified in Table 1). In this way, the smaller mechanical capacity of the GFRP is compensated by its higher fiber content. In any case, the increases in strength (with $f_{\mathrm{mc}}/f_{\mathrm{mo}}$ ratios between 2.47 and 2.74) and, above all, in ultimate strain (which increases almost 20 times with GFRP jackets), are very remarkable.

With respect to the series subjected to the action of fire, Figure 7b,c,e,f, it can be seen how the stress-strain curves exhibit the bilinear behavior described above, although the transition between the two branches occurs smoothly and a point as clearly marked as in case of undamaged series is not detected. This behavior has been reported in previous studies with concrete samples [25,26]. In terms of stress and strain at failure, it can be seen that the results are very similar to those of the equivalent series without heat treatment, with differences below 5% in all cases. Therefore, it can be concluded that the behavior of the confined specimen seems to depend basically on the properties of the FRP wrapping and not on the degree of damage that the stone core may have suffered. In relative terms, comparing the failures with those of the equivalent unconfined series, it can be seen that the ratio $f_{\mathrm{mc}}/f_{\mathrm{mo}}$ increases in the quick cooled samples, the most damaged ones, while the ratio $\varepsilon_{\mathrm{mc}}/\varepsilon_{\mathrm{mo}}$ results in the same order, regardless of the cooling system.

The great difference that has been observed between the different series tested has to do with the modulus of elasticity of the confined specimens. In the case of the undamaged stone, the slope of the initial branch of the bilinear diagram approximately coincides with that of the unconfined stone. This would mean that the jacket remains in a passive state at reduced levels of stress and that the material basically behaves as if it was not confined. However, in the series subjected to real fire, a clear improvement in the modulus of elasticity of the rock is observed from the very moment the specimen is loaded. In general terms, the modulus of elasticity of confined damaged samples is doubled with respect to the unconfined damaged ones, as the ratio $E_{\mathrm{mc}}/E_{\mathrm{mo}}$ show, and no significant differences being observed in relation to the cooling method used.

From the point of view of failure modes, Figure 8 shows some examples of the different series tested. In general, there are no substantial differences between undamaged stone or those subjected to the action of fire. Failure is determined by the tensile rupture of the FRP wrapping, producing a brittle and explosive rupture, more violent in case of CFRP confinement. FRP sheet presents break lines which follow the direction perpendicular to the fibers and, although they are usually concentrated in the midheight of the specimen, in some cases they extend to the ends and the jacket suffers a complete breakage. The stone shows a strong degradation, cracking in approximately conical shape at both ends of the specimen, while the rest of the core appears completely disintegrated. In most cases, further degradation is observed with CFRP confinement, while the samples remain somewhat more intact with the GFRP wrappings. A thin layer of stone material can always be seen adhering perfectly to the inner side of the FRP sheet, which proves the high adhesion capacity of the epoxy resin. On the other hand, it is important to note that in no case did the collapse occur in the area of the fabric overlap, which may prove that the length of the overlap adopted (25% of the perimeter of the specimen, as described previously) was sufficient in all the samples tested. Finally, it should be note that, despite their sudden and explosive nature, breakages are generally predictable. In the final stages of the test, different noises can be heard as a consequence of the cracking of the stone core or the occasional breakage of any of the FRP fibers. Furthermore, in case of GFRP jackets (has not been observed with CFRP wrapping), some white patches appear at about 70–80% of the ultimate load. These patches are attributed to a plastic flow of the resin before the FRP failure and have also been observed in other previous studies with both stone and concrete specimens.

### 3.3. Comparison with Design Guidelines

The principal guidelines for the design and construction of FRP systems for strengthening existing structures are, at present, the North American ACI 440.2R-17 [32], the British TR-55 [33], the Italian CNR-DT 200 R1/2013 [34] and the FIB Bulletin No. 14 [35]. These standards have been developed almost exclusively for the reinforcement of concrete structures, and only the Italian guide includes specific considerations for masonry and stone elements. In this section, the results obtained experimentally will be compared with the predictions of the confinement model of CNR-DT 200 R1/2013, in order to be able to assess its level of accuracy.

The compressive strength of confined stone with FRP jacketing ($f_{\mathrm{mc}}$) can be obtained from that of the unconfined stone ($f_{\mathrm{mo}}$), using Equation (1):(1)$$f_{\mathrm{mc}}{=f}_{\mathrm{mo}}\left[1+k’{\left(\frac{f_{l,\mathrm{eff}}}{f_{\mathrm{mo}}}\right)}^{\alpha_{1}}\right]$$
where $f_{l,\mathrm{eff}}$ represents the effective confinement pressure; $\alpha_{1}$ is a coefficient equal to 0.5, if no experimental data are available for its calibration; k′ is a non-dimensional coefficient that can be calculated with Equation (2):(2)$${k\prime =\alpha }_{2}{\left(\frac{g_{m}}{1000}\right)}^{\alpha_{3}}$$
where $g_{m}$ is the stone mass-density (kg/m^3^), and $\alpha_{2}$ and $\alpha_{3}$ are coefficients equal to 1, if further experimental data is not available.

The effective confinement pressure is a function of cross-sectional shape an also on the FRP arrangement; it should be calculated with Equation (3):(3)$$f_{l,\mathrm{eff}}{=k}_{\mathrm{eff}}\;f_{l}$$
where $k_{\mathrm{eff}}$ is a coefficient of efficiency equal to 1 for the specimens such as the ones that have been used in these tests (cylindrical stone specimens with FRP continuous wrapping), and $f_{l}$ is the lateral confining pressure, which may be obtained via equilibrium with Equation (4):(4)$$f_{l}=\frac{{2\;t}_{f}\;E_{f}\;\varepsilon_{f,\mathrm{red}}}{D}$$
where $t_{f}$ is the FRP thickness; $E_{f}$ is the modulus of elasticity of the FRP; D is the diameter of the cylindrical specimen; $\varepsilon_{f,\mathrm{red}}$ represents the reduced value of the FRP strain measured at collapse, to be calculated with Equation (5):(5)$$\varepsilon_{f,\mathrm{red}}=\min\left\{\frac{\eta_{a}\varepsilon_{\mathrm{fk}}}{\gamma_{f}};0.004\right\}$$
where $\eta_{a}$ is the environmental conversion factor (for indoor elements, this coefficient can be taken as 0.95 or 0.75, for CFRP or GFRP, respectively); $\varepsilon_{\mathrm{fk}}$ is the ultimate strain of FRP, obtained in direct tensile tests with plane laminates; $\gamma_{f}$ is an additional partial factor equal to 1.10 that the guide sets for the confinement model; the limited strain 0.004 is a conventional value adopted by the standard, in order to avoid an excessive damage in the confined core which could compromise the structural stability, especially under out-of-plane loads.

Table 5 shows the results obtained experimentally for the different series tested, together with the respective predictions of the confinement model adopted in CNR-DT 200 R1/2013 [34]. The values of coefficients $\alpha_{1}$, $\alpha_{2}$ and $\alpha_{3}$ specified in the standard have been maintained in the calculations. For the FRP data ($t_{f}$, $E_{f}$ and $\varepsilon_{\mathrm{fk}}$), the experimental results obtained in tensile tests are adopted (Table 1). Regarding $\varepsilon_{f,\mathrm{red}}$, the 0.004 limitation established in Equation (5) has not been considered, since it constitutes a conventional strain limit defined by the standard for the design of the reinforcement. Finally, it should be noted that, when considering the values indicated above, the ratio $\eta_{a}/\gamma_{f}$ is equal to 0.86 (CFRP) and 0.68 (GFRP). These values represent the FRP strain efficiency factor ($k_{\varepsilon }$), which has been described in previous sections. It can be seen (Table 4) how the results determined experimentally offer a good level of approach in series 03, 06, and 08, although in series 05 the value of $k_{\varepsilon }$ is 13% higher, and, in series, 09 it is 31% lower. In any case, in the predictions reported in Table 5, the coefficients proposed by the standard have been used.

It is clear that the confinement model of CNR-DT 200 R1/2013 provides a good approximation for undamaged stone series, i.e., the stone samples that were not exposed to fire. The differences from experimental results are less than 5% and with a very low dispersion index, as can be clearly seen in the comparative graph in Figure 9. Nevertheless, in case of the series exposed to high temperature, significant deviations and higher dispersions have been obtained. In these series, the standard provides very conservative results, about half of the experimental results, and it can be seen how the differences increase in the series with quick cooling, which are the most damaged. Therefore, in a first analysis, it can be concluded that for a material so seriously affected by fire, the confinement model of the Italian standard does not match the experimental results, although this discrepancy remains on the safety side.

The prediction of the standard of the strengthening ratio of Table 5 can be easily obtained with Equation (1), which can be rewritten as:(6)$$\frac{f_{\mathrm{mc}}}{f_{\mathrm{mo}}}=\left[1+k\prime {\left(\frac{f_{l,\mathrm{eff}}}{f_{\mathrm{mo}}}\right)}^{\alpha_{1}}\right]$$

The parameters k’ and $\alpha_{1}$ presented almost a constant value in the specimens tested in this research. Hence, the reinforcement ratio depended on the strength of the base material, which in this case was lower after fire exposure. If similar confinement is considered, weaker materials will present higher strength gains. However, the effective confinement pressure is limited in the code by the reduced strain value in Equation (5). This limitation is the minimum value of two possible situations: first, the failure of the FRP (considering the efficiency factor) and second, a maximum limit to control the damage in the base material. This limitation intends to avoid an excessive damage in the support, especially under transverse forces, which could compromise the lateral stability of the structure. The experimental results and predictions in Table 5 showed that, as the material was weakened, the expected strength gain prescribed in the Italian code tended to a value of 4 times. This value will correspond to a situation of perfect confinement in which the stiffness of the confinement is much higher than that of the reinforced material. For example, in a triaxial stress state, and considering the Drucker Prager yield criterion, if a confinement pressure is applied in order to totally prevent lateral strains, the longitudinal strength will be four times the unconfined strength of the material. The samples tested in this work presented strength increases up to 8.7 times the unreinforced material’s strength. Therefore, despite the FRP gave additional strength, the heated samples were expected to suffer high degradation of the core, as shown in Figure 8, which is actually what the standard intends to avoid.

## 4. Conclusions

Finally, the main conclusions derived from this research are summarized as follows:The action of fire on stone materials can seriously affect their mechanical properties. In this work, samples of calcarenite exposed to real fire with different cooling solutions (slow and quick, by spraying with cold water) have been studied, in order to evaluate the influence of the extinguishing method in case of a possible fire in a building. Compared to the properties of the undamaged stone, the results show large decreases in compressive strength (65% and 70%, with slow and quick cooling, respectively). Regarding the modulus of elasticity, very significant reductions were obtained (over 90%), regardless the cooling method used.The confinement of undamaged stone samples with FRP jacketing provides significant increases in strength (about 2.5 to 2.75 times, with GFRP and CFRP, respectively) and especially in the ultimate strain (which increases almost 20 times with glass fiber wrappings). With respect to the samples exposed to real fire, a very similar behavior is observed after retrofitting, regardless of the heat treatment and the cooling method used. In general, the response of the confined specimens depends mainly on the properties of the FRP sheet and not on the degree of damage that the stone core may have suffered.The experimental results have been compared with the predictions of the confinement model adopted in the Italian standard CNR-DT 200 R1/2013, as it is the only one of the available design guides that includes specific considerations for masonry and stone elements. This model provides an excellent approach for undamaged stone samples. However, in the series exposed to the action of fire, the values are very conservative, about half of the experimental results, and the differences increase in case of samples cooled quickly, which are more damaged. Hence, the confinement model of the Italian guideline shows notable deviations when faced with such a damaged material, although it may still be on the safety side.

## Figures and Tables

**Figure 1 polymers-12-02367-f001:**
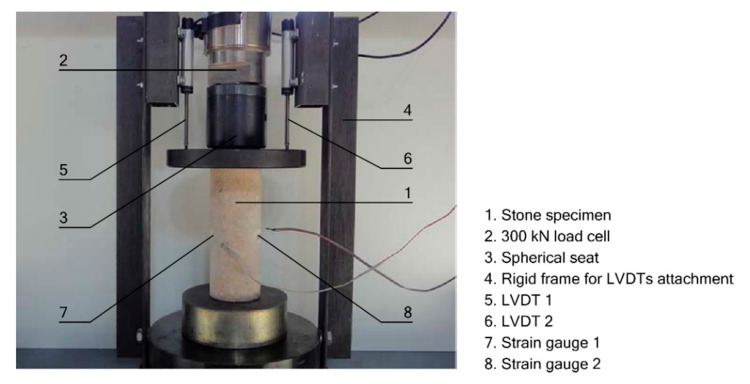
Test setup: longitudinal deformations monitored with LVDT and transverse strains with strain gauges.

**Figure 2 polymers-12-02367-f002:**
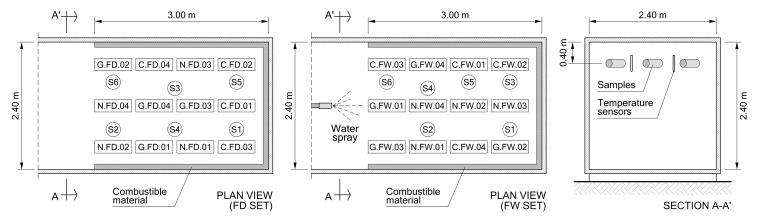
Cabin layout, showing the placement of stone samples and temperature sensors for both tests (FD (real fire exposure with slow dry cooling) and FW (real fire exposure and quick cooling by means of water spraying)).

**Figure 3 polymers-12-02367-f003:**
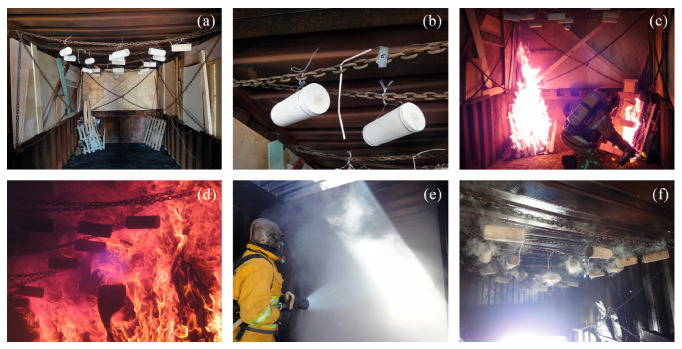
Real fire tests: (**a**) general view of the container; (**b**) detail of samples and thermocouples attachment; (**c**) ignition of combustible material; (**d**) fully developed fire; (**e**) fire extinction; (**f**) view of specimens after being cooled.

**Figure 4 polymers-12-02367-f004:**
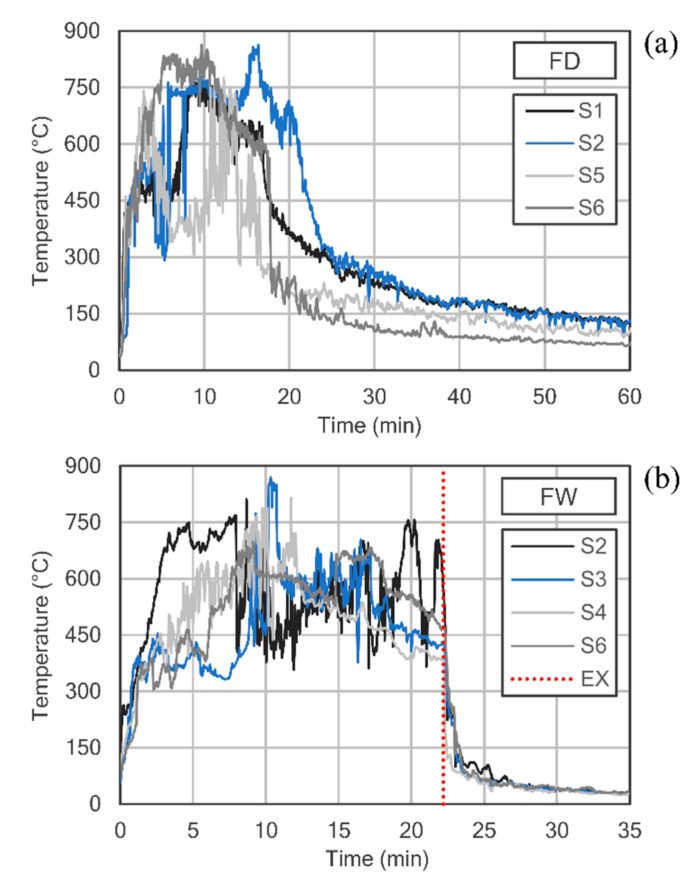
Temperature records inside the cabin for both tests performed: (**a**) FD and (**b**) FW.

**Figure 5 polymers-12-02367-f005:**
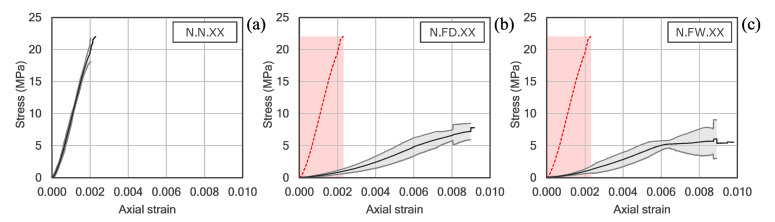
Stress-strain curves of unconfined stone samples: (**a**) undamaged, or after fire exposure with (**b**) slow air-cooling or (**c**) quick water-cooling.

**Figure 6 polymers-12-02367-f006:**
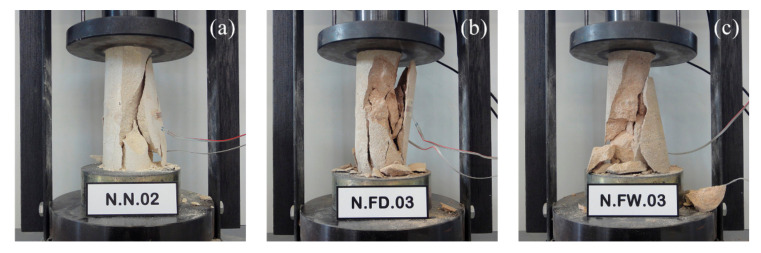
Examples of failure of unconfined stone samples: (**a**) undamaged, or after fire exposure with (**b**) slow air-cooling or (**c**) quick water-cooling.

**Figure 7 polymers-12-02367-f007:**
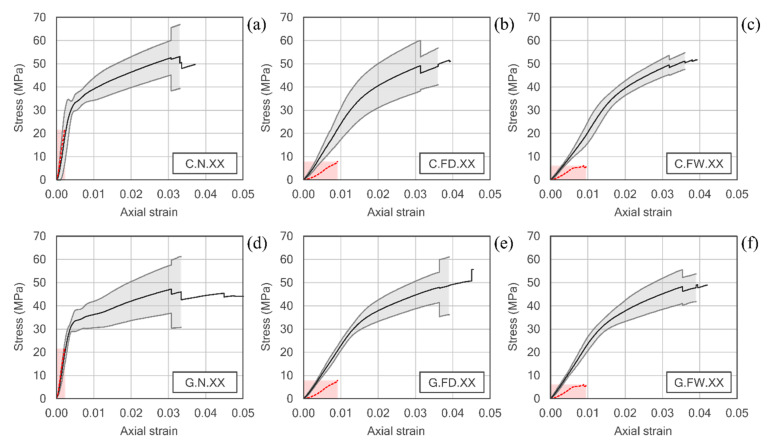
Stress-strain curves of FRP confined stone samples: CFRP, (**a**) undamaged, or after fire exposure with (**b**) slow air-cooling or (**c**) quick water-cooling; GFRP, (**d**) undamaged, or after fire exposure with (**e**) slow air-cooling or (**f**) quick water-cooling.

**Figure 8 polymers-12-02367-f008:**
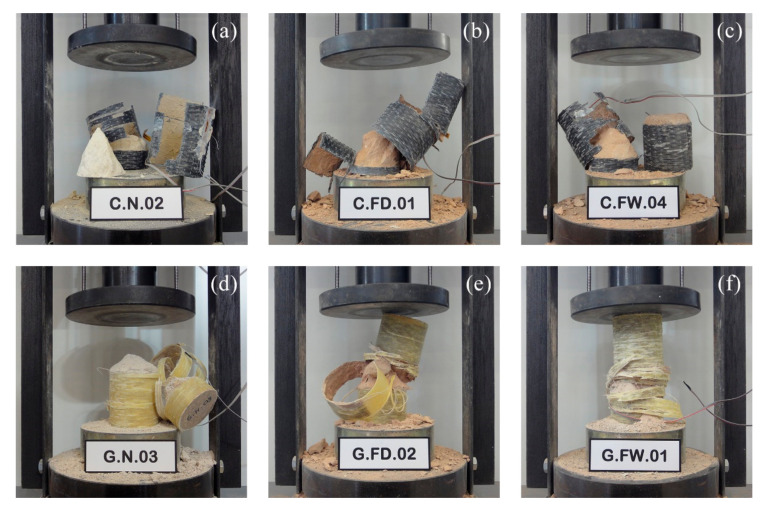
Examples of failure of FRP confined stone samples: CFRP, (**a**) undamaged, or after fire exposure with (**b**) slow air-cooling or (**c**) quick water-cooling; GFRP, (**d**) undamaged, or after fire exposure with (**e**) slow air-cooling or (**f**) quick water-cooling.

**Figure 9 polymers-12-02367-f009:**
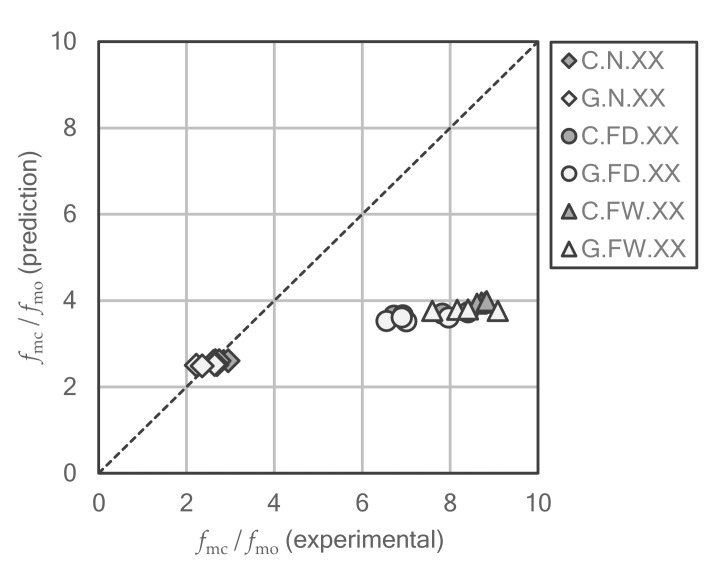
Comparison of the strengthening ratios $f_{\mathrm{mc}}/f_{\mathrm{mo}}$ obtained experimentally and predicted by the confinement model adopted in the Italian guideline for masonry or stone FRP wrapping.

**Table 1 polymers-12-02367-t001:** Main properties of raw materials and composite specimens.

	Fabric (Unidirectional) ^a^		Epoxy Resin ^a^	Composite ^b^	
	Carbon	Glass		CFRP	GFRP
Weight (g/m^2^)	300	900	-	-	-
Thickness (mm)	0.166 ^c^	0.480 ^c^	-	0.81 (12.54%) ^d^	1.08 (3.36%) ^d^
Tensile strength (MPa)	4830	2560	40	637 (6.43%)	539 (7.48%)
Elastic modulus (MPa)	230,000	80,700	1400	56,078 (7.60%)	25,344 (6.06%)
Ultimate strain (%)	2	3–4	1.8	1.16 (6.01%)	2.21 (5.15%)

^a^ Values provided by supplier. ^b^ Experimental average values from uniaxial tensile tests (coefficient of variation, in brackets). ^c^ Dry fabric thickness. ^d^ Cured sheet thickness, measured with micrometer.

**Table 2 polymers-12-02367-t002:** Identification of specimens.

Set.	Label	Confinement	Layers	Stone treatment	Samples
01	N.N.XX	-	-	-	4
02	C.N.XX	CFRP	1	-	4
03	G.N.XX	GFRP	1	-	4
04	N.FD.XX	-	-	Fire exposure, slow dry cooling	4
05	C.FD.XX	CFRP	1	Fire exposure, slow dry cooling	4
06	G.FD.XX	GFRP	1	Fire exposure, slow dry cooling	4
07	N.FW.XX	-	-	Fire exposure, quick water cooling	4
08	C.FW.XX	CFRP	1	Fire exposure, quick water cooling	4
09	G.FW.XX	GFRP	1	Fire exposure, quick water cooling	4

**Table 3 polymers-12-02367-t003:** Unconfined samples: experimental results (coefficient of variation, in brackets).

Set	Label	*g*_m_ (kg/m^3^)	*f*_mo_ (MPa)	ε_mo_	*E*_mo_ (MPa)
01	N.N.XX	1982 (1.41%)	19.99 (8.07%)	0.00212 (6.04%)	11859 (4.01%)
04	N.FD.XX	1988 (0.32%)	7.00 (8.36%)	0.00885 (5.81%)	1214 (9.69%)
07	N.FW.XX	1974 (1.98%)	6.00 (19.08%)	0.00918 (19.63%)	1162 (12.32%)

**Table 4 polymers-12-02367-t004:** Fiber reinforced polymer (FRP) confined samples: experimental results (coefficient of variation, in brackets).

Set	Label	*f*_mc_ (MPa)	*f*_mc_/*f*_mo_	ε_mc_	ε_mc_/ε_mo_	*E*_mc_ (MPa)	*E*_mc_/*E*_mo_	ε_mc,t_	ε_fu_	*k* _ε_
02	C.N.XX	54.70 (5.4%)	2.74	0.03508 (15.4%)	16.55	12236 (17.3%)	1.03	^a^	0.0116 (6.0%)	^a^
03	G.N.XX	49.47 (9.2%)	2.47	0.04086 (23.7%)	19.27	10102 (8.7%)	0.85	0.01489 (10.1%)	0.0221 (5.2%)	0.67
05	C.FD.XX	52.23 (10.6%)	7.46	0.03642 (9.8%)	4.11	2592 (17.6%)	2.13	0.01122 (8.9%)	0.0116 (6.0%)	0.97
06	G.FD.XX	49.68 (8.5%)	7.10	0.04216 (12.6%)	4.76	2384 (4.3%)	1.96	0.01541 (12.6%)	0.0221 (5.2%)	0.70
08	C.FW.XX	52.40 (1.1%)	8.73	0.03627 (9.0%)	3.95	2545 (3.3%)	2.19	0.00995 (11.70%)	0.0116 (6.0%)	0.86
09	G.FW.XX	49.84 (7.4%)	8.31	0.03910 (7.1%)	4.26	2470 (8.9%)	2.13	0.01029 (19.16%)	0.0221 (5.2%)	0.47

^a^ No data available in this set.

**Table 5 polymers-12-02367-t005:** Experimental results versus theoretical CNR-DT 200 R1/2013 prediction.

Set	Label	Experimental		Prediction		$f_{\mathrm{mc}}^{\exp}/f_{\mathrm{mc}}^{\mathrm{theo}}$
		*f*_mc_ (MPa)	*f*_mc_/*f*_mo_	*f*_mc_ (MPa)	*f*_mc_/*f*_mo_	
02	C.N.XX	54.70	2.74	51.95	2.60	1.05
03	G.N.XX	49.47	2.47	50.09	2.51	0.98
05	C.FD.XX	52.23	7.46	25.75	3.68	2.03
06	G.FD.XX	49.68	7.10	24.95	3.57	1.99
08	C.FW.XX	52.40	8.73	23.73	3.95	2.21
09	G.FW.XX	49.84	8.31	22.65	3.78	2.20

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
