# Peer review of "FRP Confinement of Stone Samples after Real Fire Exposure"

_polymers, 2020, doi:10.3390/polym12102367_

Round 1

Reviewer 1 Report

This study investigates the axial compressive behavior of FRP confined fire-damaged stone. Effects of FRP types and cooling methods are considered. It is an interesting study. This paper is suggested publishing after a minor revision.

Line 143: The gauge length of the LVDT should be provided.

Line 148: As shown in Table 1, the tensile strength and elastic modulus of the FRP composites are much smaller than those of the fabric. Why?

Line 176: The details (e.g., LVDT, load cell, spherical hinge) should be marked in Figure 1.

Line 185: The placement of the thermocouples in the FD test is different from that in the FW test. Why?

Line 189: Will the responses of the specimens be affected by their placement? For example, specimen C.FD.02. is subjected to fire on two sides, while specimen specimen C.FD.01. is only subjected to fire on one side.

Line 190: Figure. 2 is not clear enough. Please revise.

Line 294: As shown in Table 4, the FRP efficiency factor of the CFRP confined specimens is always larger than that of the GFRP confined specimens. Why?

Line 339: For specimens without fire exposure, the elastic modulus of the FRP confined stone is close to that of the unconfined stone. However, for specimens with fire exposure, the elastic modulus of the FRP confined stone is much larger than that of the unconfined stone. Why?

Author Response

We sincerely appreciate the reviewer's comments. We respond in an annexed Word document to each of the suggestions, which will undoubtedly improve the quality of the paper.

Reviewer 2 Report

The submitted paper “polymers-951650” entitled: “FRP confinement of stone samples after real fire exposure” is an original experimental study that investigates the compressive behaviour of cylindrical stone specimens that first damaged by high temperature exposure by real fire action and then retrofitted by of Carbon and Glass FRP confinement jackets. The effectiveness of the examined FRP jackets is investigated by the results of uniaxial compression tests. Experimental results are also compared with available confinement model adopted in the Italian standard CNR-DT 200 R1/2013 and useful concluding remarks have been drawn. The paper falls within the scope of the Journal and deals with a topic still open to question since the existing published work in this field of study is very limited. New quantitative and qualitative results are presented in a clear and organized manner. Manuscript is well-structured and easily understood. As an overall and based on the aforementioned comments, the article could be published after minor revision based on the following recommendations:

  1. Research significance and subsequent impact of the presented study on the state of the practice could further be highlighted in order to promote the objectives of this interesting work.
  2. Further discussion and justification of the experimentally obtained failure modes showed in Fig. 8 would be helpful.
  3. Commentary of the comparisons presented in Figure 9 is rather limited and could be enriched with further explanatory remarks.

Author Response

(The authors gave the same response as above.)
